# Guaranteed Simply Connected Mesh Reconstruction from an Unorganized Point Cloud

**Liyan Chen**[1]   **Jingyi Li**[2,3]   **Qixing Huang**[1]
[1]The University of Texas at Austin   [2]École Polytechnique   [3]Inria Saclay
{liyanc, huangqx}@cs.utexas.edu   jingyi.li@polytechnique.edu

## Abstract

We introduce an approach that reconstructs a closed surface mesh from a noisy point cloud, with a guarantee that the reconstructed surface is simply connected (homeomorphic to a 2-sphere). This task enjoys a wide range of applications, e.g., 3D organ and vessel reconstruction from CT scans. Central to our approach is a robust module that takes a collection of oriented triangles in a 3D triangulation as input and outputs a simply connected volumetric mesh whose boundary approximates the input triangles. Starting from a 3D Delaunay triangulation of the input point cloud and initial triangle orientations obtained through a spectral approach, our approach alternates between applying the module to obtain a reconstruction and using that reconstruction to reorient the input triangles. Experimental results on real and synthetic datasets demonstrate the robustness and accuracy of our approach.

## 1 Introduction

Reconstructing a 3D mesh surface from noisy point clouds is a fundamental problem in computer graphics and related domains Berger et al. (2017). Existing methods—ranging from computational geometry to implicit function and neural approaches—achieve strong geometric fidelity but generally lack topological control. In particular, no prior method guarantees a *simply connected* reconstruction, a property essential for modeling structures such as human organs and vessels (cf. CrossSDF Walker et al. (2025)). This gap stems from the absence of computational tools for effective topological control. We review these approaches in detail in Section 2.

In this paper, we introduce a mesh surface reconstruction approach that approximates the underlying surface while guaranteeing that the reconstruction is simply connected and closed, as shown in Figure 2. We present a nontrivial generalization of the winding number field paradigm Feng et al. (2023), extending topological control from 1D curves on 2D surfaces to 2D surfaces embedded in 3D. Our formulation takes as input a collection of oriented triangles from a 3D triangulation and outputs a triangle mesh that is closed, simply connected, and manifold. Motivated by IPSR Hou et al. (2022), we introduce an approach that combines an initialization phase and an alternating phase. The initialization phase computes a 3D Delaunay triangulation of the input point cloud and establishes consistent orientations of the triangles. A key contribution of this paper is a spectral method for computing these orientations. The algorithm then alternates between reconstructing a surface from the current triangle orientations and using the resulting surface to select and reorient the input triangles. This procedure is robust and effectively removes outlier triangles from the input.

We evaluated our approach on the 3D medical reconstruction benchmark introduced in CrossSDF Walker et al. (2025), as well as on a synthetic benchmark with controlled noise levels and sampling densities. The experiments demonstrate that our method is the only one that guarantees simply connected reconstructions while matching state-of-the-art methods in geometric accuracy. An ablation study further justifies the design choices underlying our approach.

## 2 Related Work

Existing surface reconstruction approaches can be broadly divided into two categories. The first category consists of computational geometry approaches Amenta et al. (1998); Cheng et al. (2013).

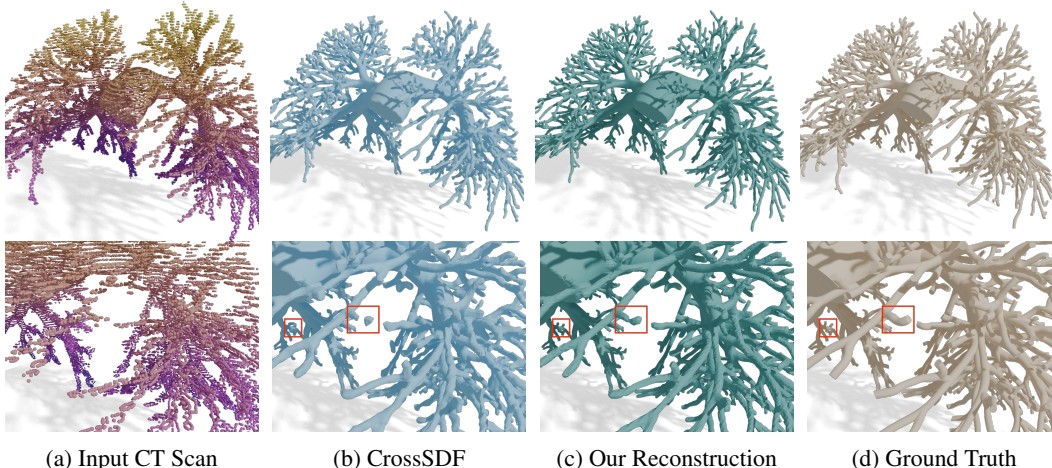

| (a) Input CT Scan | (b) CrossSDF | (c) Our Reconstruction | (d) Ground Truth |

Figure 1: Visual comparison of pulmonary vascular reconstruction. The top row displays the full rendered results, while the bottom row presents corresponding zoomed-in views. From left to right: (a) the input CT scan; (b) the reconstruction of Walker et al. (2025) with **6 connected components** and numerous spurious handles; (c) our simply connected reconstruction; and (d) the simply connected ground truth.

The advantage of these approaches is that they offer theoretical guarantees when points are clean and their distributions are adaptive to the curvature of the underlying shape. However, their performance degrades significantly in the presence of outliers in the input data Dey & Goswami (2004). Our method belongs to this category but remains robust to noisy input. The work most closely related to ours is the spectral approach of Kolluri et al. (2004), which, unlike ours, does not guarantee the reconstruction of a simply connected surface mesh.

Another category of surface reconstruction methods first fits an implicit function to the input points and then extracts a level-set using an algorithm like Marching Cubes Lorensen & Cline (1987). Among classic techniques, Poisson Surface Reconstruction (PSR) Kazhdan et al. (2006) remains highly influential but, like many early methods Hoppe et al. (1992); Carr et al. (2001), requires oriented normals as input. This requirement extends to many modern deep learning approaches such as DeepSDF Park et al. (2019) and SIREN Sitzmann et al. (2020). While methods such as IPSR Hou et al. (2022), SAL Atzmon & Lipman (2020), and SALD Atzmon & Lipman (2021) have been developed to handle unoriented point clouds, none provides explicit control over the topology of the reconstructed surface, which is the primary focus of this paper.

Our approach is inspired by the winding number field Jacobson et al. (2013), a concept recently used to guide normal orientation in iterative frameworks Xu et al. (2023); Liu et al. (2025). Our work builds most directly on Feng et al. (2023), which introduces a Helmholtz-Hodge Decomposition (HHD) of a discrete winding number field to capture the non-trivial topology of curves on a surface. In this paper, we generalize this framework from 2D curves to 3D surfaces and address several critical challenges. First, since the HHD framework requires a consistently oriented triangulation as input, we propose a method to initialize triangle orientations spectrally and then refine them in an alternating scheme alongside the surface reconstruction, akin to IPSR Hou et al. (2022). Second, we employ robust optimization to suppress spurious topological handles, which naturally promotes outlier removal and enforces a desired simple topology.

## 3 OVERVIEW

We begin with the problem statement in Section 3.1, followed by an overview of our approach in Section 3.3.

### 3.1 PROBLEM STATEMENT

The input is a collection of $n$ points $\mathcal{P} = \{\boldsymbol{p}_i \in \mathbb{R}^3\}$ sampled from some underlying 3D object. Typical examples include data acquired from 3D sensors such as LiDAR, depth cameras, and CT

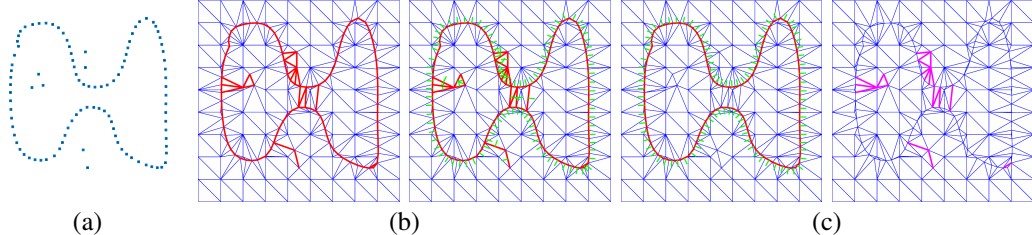

(a)        (b)        (c)

Figure 2: Overview of our approach. (a) Input unoriented point cloud. (b) The initialization stage computes a base triangulation (left) and an initial orientation of the boundary faces $\mathcal{F}_0$. (c) The alternating stage iterates between computing the correction surface $\Lambda$ and updating the current reconstruction, represented as a collection of oriented faces.

scans. Such point sets are often irregularly distributed and may contain outliers. The output is a closed triangle mesh $\mathcal{M}_2 = (\mathcal{V}, \mathcal{E}, \mathcal{F})$ where $\mathcal{V}$ partially overlaps with $\mathcal{P}^1$, and $\mathcal{M}_2$ is simply connected — that is, path-connected and free of nontrivial loops. In other words, $\mathcal{V}$ excludes outliers present in $\mathcal{P}$ and introduces additional points to satisfy the simply connected topological constraint and fill missing regions.

### 3.2 ALGEBRAIC FORMULATION FOR TOPOLOGICAL GUARANTEES

A core contribution of our method is the provision of a global algebraic guarantee for a simply connected reconstruction, effectively shifting the topological constraint from a complex combinatorial domain into an efficient linear domain. We formally derive this guarantee using De Rham's Theorem and the Hodge Isomorphism Theorem Warner (1983) (see Appendix B.3 for the further statements).

The theoretical mechanism relies on the Helmholtz-Hodge Decomposition (HHD), which separates an input 1-form into exact, co-exact, and harmonic components. Crucially, the Hodge Isomorphism Theorem states that for a smooth manifold, the space of harmonic 1-forms is strictly isomorphic to the first cohomology group $H^1$, which is the algebraic generator of topological handles. By algorithmically removing these harmonic 1-forms via a linear solve, our formulation strictly enforces a trivial $H^1$. The final surface, constructed as the level set of this optimized field, is therefore guaranteed to be simply connected by construction rather than by probabilistic or heuristic approximation.

### 3.3 APPROACH OVERVIEW

Figure 2 illustrates our reconstruction pipeline, which consists of an initialization stage and an alternating reconstruction stage. The initialization stage first computes a 3D Delaunay triangulation $\overline{\mathcal{M}}_3 = (\overline{\mathcal{V}}, \overline{\mathcal{E}}, \overline{\mathcal{F}}, \overline{\mathcal{T}})$ together with a subset of oriented faces $\mathcal{F}_0 \subset \overline{\mathcal{F}}$ from the input points. Each face in $\mathcal{F}_0$ is assigned an orientation value in $\{-1, 1\}$. To avoid elongated tetrahedra, additional points are inserted into the triangulation under the guidance of an octree built from the input. The subset $\mathcal{F}_0$ is then defined as the set of tetrahedral faces whose vertices belong to the input points. We then assign an orientation to each face in $\mathcal{F}_0$ using a spectral formulation. This initialization stage is described in Section 5.

In the alternating reconstruction stage, we first apply the core module (see Section 4). This module takes as input the 3D triangulation $\overline{\mathcal{M}}_3$ together with the oriented faces $\mathcal{F}_0$ defined above, and outputs a closed, simply connected mesh whose faces $\mathcal{F}$ approximate $\mathcal{F}_0$. The resulting mesh $\mathcal{M}_2$ then induces an optimized set of oriented faces $\mathcal{F}^\star \subset \overline{\mathcal{F}}$. We update $\mathcal{F}_0 \leftarrow \mathcal{F}_0 \cap \mathcal{F}^\star$ and reapply the core module. This alternating process terminates once $\mathcal{F}^\star$ stabilizes, typically within 1-4 iterations.

## 4 CORE MODULE

We begin with an overview of the core module in Section 4.1, which generalizes the winding number field approach of Feng et al. (2023), and then describe its details in Section 4.2.

---

[1]We assume that input inliers are accurate, a setting that applies to many sensors. See Section 7 for further discussion.

## 4.1 MODULE OVERVIEW

The foundation of our approach is the winding number field $u : \mathbb{R}^3 \rightarrow \mathbb{Z}_{\geqslant 0}$ associated with an oriented 2D surface $S$ embedded in $\mathbb{R}^3$, where $u(\boldsymbol{p})$ denotes the number of times $S$ encloses a point $\boldsymbol{p} \in \mathbb{R}^3$. It is clear that $u$ is piecewise constant with discontinuities on the surface $S$.

A key property of $u$ is that it encodes all topological features of $S$ through the harmonic component of its Hodge-Helmholtz decomposition (HHD). To exact this information, we first take the discrete 1-form $\boldsymbol{\omega} = Du$, where $D$ denotes the discrete gradient operator mapping the vertex-based scalar field $u$ to a 1-form defined on edges. Then the Hodge-Helmhotz decomposition states that any discrete 1-form $\boldsymbol{\omega}$ can be uniquely and orthogonally decomposed into three components: an exact part, a coexact part and a harmonic part:

$$\boldsymbol{\omega} = d_0 \boldsymbol{\alpha} + \delta_2 \boldsymbol{\beta} + \boldsymbol{\gamma}. \tag{1}$$

Here, $d_0$ is the discrete gradient operator and $\delta_2$ is the discrete curl operator (the adjoint of the 2-form exterior derivative $d_1$). The component $d_0 \boldsymbol{\alpha}$ is curl-free, $\delta_2 \boldsymbol{\beta}$ is divergence-free, and the harmonic component $\boldsymbol{\gamma}$ is both curl-free and divergence-free. The harmonic part $\boldsymbol{\gamma}$ represents the de Rham cohomology class of $\boldsymbol{\omega}$, which is isomorphic to the non-trivial 1-homology of the domain. In our context, $\boldsymbol{\gamma}$ serves as the dual representation of all nonbounding 2-cycles implicitly defined by the input $\mathcal{F}_0$. A more formal introduction of HHD and the associated discrete operators is provided in Appendix A and Appendix B.

Another attractive property of $u$ is that it can be recovered, up to a global constant, by evaluating its jump across the surface $S$. However, this recovery procedure implicitly assumes that the inside-outside labeling at each point of $S$ is known, so that the absolute 'jump' Feng et al. (2023) in $u$ cross $S$ can be properly defined. Such orientation information is usually unavailable from an unorganized point cloud, leading to the well-known consistent normal orientation problem. In Section 5.2, we will introduce how to compute such orientation information.

## 4.2 MODULE PIPELINE

The first step in discretizing the winding number field $u$ on the tetrahedral mesh $\overline{\mathcal{M}}_3$ is to determine an appropriate representation. A natural approach is to model $u$ as a piecewise-constant function that assigns a constant value to each tetrahedron $T$ and enforces jump constraints of the form $u^T - u^{T'}$ across shared faces of adjacent tetrahedra. However, this formulation leads to an underconstrained system and does not offer an effective way to extract topological signatures. Motivated by Feng et al. (2023), we instead represent $u$ in a reduced coordinate system by defining $u_v^T := u^T(v)$ for each vertex $v$, and expressing

$$u_v^T = (u_0)_v + c_v^T, \tag{2}$$

where $u_0$ denotes an underlying vertex-based field, and $c$ encodes the jump information across $\mathcal{F}_0$. Specifically, $c_v^T$ is computed by accumulating local jump updates along a path of face-adjacent tetrahedra from a local reference tetrahedron $T_{\text{ref}}$ (where $c \equiv 0$) to $T$. Consequently, $c_v^T$ is constant within each tetrahedron; that is, $c_v^T = c_{v'}^T$ for any vertices $v, v'$ of $T$.

Consider two adjacent tetrahedra $T_1$ and $T_2$ sharing a face $f = (i, j, k)$. For any vertex $v \in \{i, j, k\}$, the difference of the potential $u$ can be written as

$$u_v^{T_1} - u_v^{T_2} = ((u_0)_v + c_v^{T_1}) - ((u_0)_v + c_v^{T_2}) = c_v^{T_1} - c_v^{T_2}. \tag{3}$$

Since the base field $(u_0)_v$ cancels, this difference is independent of the vertex $v$ and is therefore constant across the face $f$. By construction, the path-integral definition of $c$ enforces

$$c_v^{T_1} - c_v^{T_2} = \Gamma_f \in \{-1, 1\},$$

where the sign depends on the orientation of $f$. Hence, this parameterization satisfies the prescribed jump conditions.

We define the 1-form $\overline{\omega}$ derived from the Darboux derivative of $u_v^T$ as follows. For each edge $(i, j)$,

$$\overline{\omega}_{ij} = \text{Avg}_{T_p \in \text{fan}(ij)}(u_j^{T_p} - u_i^{T_p}) = (u_i - u_j) + \omega_{ij}, \tag{4}$$

where

$$\omega_{ij} = \text{Avg}_{T_p \in \text{fan}(ij)}(c_j^{T_p} - c_i^{T_p}),$$

and $\text{fan}(ij)$ denotes the set of all tetrahedra incident to the edge $(i, j)$.

Our approach is based on the HHD of the 1-form $\overline{\omega}$. Since $\overline{\omega}$ and $\omega$ differ only by the exact form $du$ (the Darboux derivative of $u$), their harmonic components coincide. Therefore, the harmonic component can be computed from $\omega$, which can be directly obtained from the input.

### 4.2.1 COMPUTATION OF HELMHOLTZ-HODGE DECOMPOSITION

In the HHD formulation shown in Equation (1), the operators $d_i, \delta_i$ are presented as sparse matrices, while $\omega, \alpha, \beta$ and $\gamma$ are expressed in vector form. We next describe how to compute $\alpha$ and $\beta$, both of which require solving linear systems.

Specifically, the exact component $\alpha$ is given by the linear system

$$L_0 \alpha = \delta_1 \omega, \tag{5}$$

where $L_0$ is the Dihedral Angle Laplacian (DAL), and $\delta_1$ is the discrete divergence operator.

The weight of an edge $(i, j)$ in the Dihedral Angle Laplacian $L_0$ is defined by the sum of contributions from all tetrahedra incident on that edge:

$$w_{ij} = \sum_{T \in \text{fan}(ij)} w_{ij}^T, \tag{6}$$

where the contribution $w_{ij}^T$ from a single tetrahedron is determined by its dihedral angle geometry (see Appendix C). With these weights, the DAL takes the form

$$L_0 = E^\top \text{diag}(w_{ij}) E,$$

where $E \in \{-1, 0, 1\}^{|\mathcal{E}| \times \mathcal{V}}$ is the matrix representation of the discrete gradient operator $d_0$. Similarly, the coexact component $\beta$ is obtained by solving the linear system

$$L_2 \beta = d_1 \omega,$$

where $d_1$ is the discrete curl operator and

$$L_2 = F \text{diag}(w_{ij})^{-1} F^\top$$

is the matrix representation of the 2-form Laplacian. Here, $F \in \{-1, 0, 1\}^{|\mathcal{F}| \times |\mathcal{E}|}$ encodes the operator $d_1$.

The harmonic 1-form is given by

$$\gamma = \omega - d_0 \alpha - \delta_2 \beta'$$

In the special case $\gamma = du$, the decomposition yields $\alpha = u, \beta = 0, \gamma = 0$. Because the HHD is linear, $\overline{\omega}$ and $\omega$ share the same harmonic component.

### 4.2.2 CORRECTION 2-CHAIN CONSTRUCTION

After extracting $\gamma$, which encodes the dual representation of all nonbounding topological features, we construct a correction 2-chain $\Lambda$ that captures these features. The chain $\Lambda$ identifies the faces to be added to or removed from $\mathcal{F}_0$ in order to obtain a simply connected surface.

One challenge is that the harmonic form $\gamma$ determines only a homology class for the correction surface $\Lambda$. Indeed, subtracting the boundary of any 3-chain $W$ yields another valid surface $\Lambda - \partial_3 W$, since $\Lambda$ and $\Lambda - \partial_3 W$ are homologous. To resolve this ambiguity, we seek a correction surface $\Lambda$ that is consistent with $\gamma$ while minimizing a suitable objective function.

To this end, we parameterize $\Lambda$ using per-tetrahedron potentials $\sigma_T$ for $T \in \mathcal{T}$. For a face shared by two adjacent tetrahedra $T_1$ and $T_2$, we define

$$\Lambda_{T_1 T_2} := \sigma_{T_1} - \sigma_{T_2}.$$

For any edge $(i, j) \in \mathcal{E}$, the consistency constraint is given by

$$(D\sigma)_{ij} = \gamma_{ij}, \tag{7}$$

where $D$ is the Darboux gradient operator.

To determine the correction surface, we promote sparsity of $\Lambda$ while accounting for the geometric importance of each face through its area. Unlike Feng et al. (2023), which formulates this problem as a linear program, we use the $L^0$ norm for minimization:

$$\min_{\sigma_T} \sum_{f \in \mathcal{F}} \text{Area}(f)|\Lambda_f|^0. \tag{8}$$

This formulation is motivated by two considerations. First, the $L^0$ penalty promotes sparse corrections and yields improved robustness in the presence of outliers in $\mathcal{F}_0$. Second, it admits an efficient solution via iteratively reweighted least squares, which scales more favorably to large problems than solving a linear program. Following this strategy, we approximate Equation (8) by the weighted quadratic objective

$$\min_{\sigma_T} \sum_{f \in \mathcal{F}} \text{Area}(f)w_f \Lambda_f^2, \tag{9}$$

and solve the linearly constrained quadratic program defined by Equation (7) to obtain an updated solution $\sigma_T^\star$. Note that for an interior face $f$ shared by tetrahedra $T_1$ and $T_2$, we use the equivalent notations $w_f$ and $w_{T_1 T_2}$. The weights are then updated according to

$$w_{T_1 T_2} = \frac{\epsilon^2}{\epsilon^2 + (\sigma_{T_1}^\star - \sigma_{T_2}^\star)^2},$$

where $\epsilon$ is a small positive constant introduced to avoid division by zero. This procedure is repeated until $\sigma_T^\star$ converges.

### 4.2.3 Final Surface Reconstruction

Let $\Gamma_{\mathcal{F}_0}$ denote the 2-chain defined by $\mathcal{F}_0$. After computing the correction 2-chain $\Lambda$, we construct a "clean" 2-chain $\Gamma' = \Gamma_{\mathcal{F}_0} - \Lambda$, which removes the nonbounding components of $\Gamma_{\mathcal{F}_0}$. Because $\Gamma'$ may introduce new faces that bridge previously disconnected components, we improve the smoothness of the final reconstruction by solving the Laplacian system once more:

$$\mathbf{L}\mathbf{w}_0 = \mathbf{b}', \tag{10}$$

where $\mathbf{b}'$ is the source vector derived from $\Gamma'$. The resulting function

$$\mathbf{w} = \mathbf{w}_0 + c \tag{11}$$

is a jump harmonic field whose discontinuities correspond only to homologically trivial surfaces.

The final reconstruction is obtained by rounding the values of $\mathbf{w}$ within each tetrahedron to produce an integer-valued 3-chain $W$. The reconstructed surface $S$ is then defined as the boundary of $W$:

$$S = \partial_3 W. \tag{12}$$

By construction, $S$ is the boundary of a connected volume and therefore forms a closed 2-chain that is homologous to zero, guaranteeing that $S$ has simply connected topology.

## 5 Initialization Stage

This section presents the initialization stage of our approach. In Section 5.1, we describe how to obtain a tetrahedral mesh $\overline{\mathcal{M}}_3 = (\overline{\mathcal{V}}, \overline{\mathcal{E}}, \overline{\mathcal{F}}, \overline{\mathcal{T}})$ where $\mathcal{P} \subseteq \overline{\mathcal{V}}$. In Section 5.2, we present a key contribution of this paper, a spectral approach that computes consistent oriented faces from $\mathcal{P}$.

### 5.1 Delaunay Triangulation

A simple initialization approach is to compute the 3D Delaunay triangulation of $\mathcal{P}$ to obtain $\overline{\mathcal{M}}_3$. However, this approach may introduce tetrahedra with very long edges and fails to provide tetrahedra on both sides of the underlying boundary triangles, a property that is critical for our framework. Motivated by Kolluri et al. (2004), we therefore augment the input point set $\mathcal{P}$ with the cell centers of a sparse grid enclosing $\mathcal{P}$, and perform 3D Delaunay triangulation on the augmented set to construct the tetrahedral mesh $\overline{\mathcal{M}}_3$ used in our algorithm.

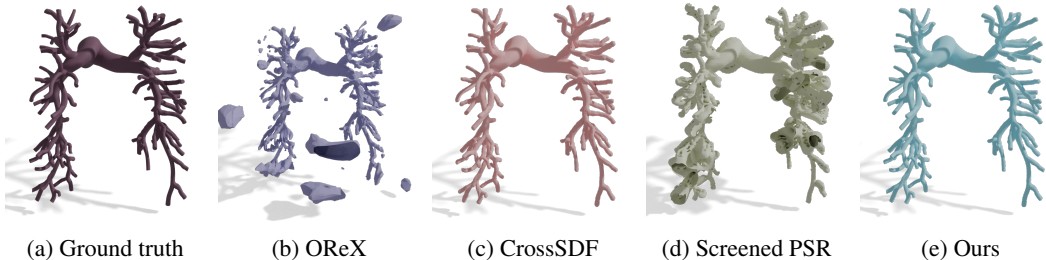

| (a) Ground truth | (b) OReX | (c) CrossSDF | (d) Screened PSR | (e) Ours |

Figure 3: Qualitative comparisons between our approach and baseline approaches.

## 5.2 TRIANGLE ORIENTATION

Let $\mathcal{F}_0 \subset \mathcal{F}$ denote the set of faces in $\overline{\mathcal{M}}_3$ whose three vertices lie in $\mathcal{P}$. Our goal is to determine a consistent orientation of each face $f \in \mathcal{F}_0$ for subsequent optimization. To this end, we associate with each face $f \in \mathcal{F}_0$ an indicator variable $x_f \in \{-1, 1\}$, and collect these variables into a vector $\boldsymbol{x}$. We then optimize $\boldsymbol{x}$ using the formulation derived from the framework in Section 4. At an abstract level, the harmonic component can be written as

$$\boldsymbol{\gamma} = A\boldsymbol{x}, \tag{13}$$

where $A$ is a linear operator determined by the coefficients $c^T$, the Darboux derivative operator, and the Hodge-Helmholtz decomposition. We seek an orientation $\boldsymbol{x}$ such that $\boldsymbol{\gamma}$ vanishes.

A technical challenge is that $\boldsymbol{x}$ is an integer vector, making direct minimization of $\|A\boldsymbol{x}\|$ infeasible. To address this, we relax $\boldsymbol{x}$ to be a real-valued vector and minimize the ratio $\frac{\|A\boldsymbol{x}\|^2}{\|\boldsymbol{x}\|^2}$, whose optimal solution $\boldsymbol{x}^\star = \boldsymbol{u}_1(A^\top A)$ is the eigenvector associated with the smallest eigenvalue of $A^\top A$. We then recover a discrete orientation by rounding $\boldsymbol{x}^\star$ to $\{-1, 1\}^{|\mathcal{F}_0|}$ according to the sign of each entry. In the following, we describe the construction of $A$ and the computation of the smallest-eigenvalue eigenvector of $A^\top A$.

First, note that each $c^T$ depends on $\boldsymbol{x}$. Therefore, we can encode

$$\boldsymbol{\omega} = B\boldsymbol{x}, \tag{14}$$

where $B \in \mathbb{R}^{|\overline{\mathcal{E}}| \times |\mathcal{F}_0|}$ is a sparse matrix. Based on Equation (1), we have

$$\begin{aligned}\boldsymbol{\gamma} =& \boldsymbol{\omega} - d_0\boldsymbol{\alpha} - \delta_2\boldsymbol{\beta} \\ =& (I - d_0 L^\dagger \delta_1 - \delta_2 L_2^\dagger d_1)\boldsymbol{\omega},\end{aligned} \tag{15}$$

where $L^\dagger$ and $L_2^\dagger$ denote the Moore-Penrose pseudoinverses of the 0-form and 2-form Laplacians. Combining Equation (13), Equation (14) and Equation (15), we have

$$A = (I - d_0 L^\dagger \delta_1 - \delta_2 L_2^\dagger d_1)B. \tag{16}$$

We use the Lanczos algorithm to compute the eigenvector associated with the smallest eigenvalue of $A^\top A$. Each iteration requires solving six sparse linear systems (four involving $L$ and two involving $L_2$).

## 6 EXPERIMENTAL RESULTS

This section presents an ablation study of our approach. We begin with the experimental setup in Section 6.1. We then present results on real and synthetic datasets in Section 6.2 and Section 6.3, respectively. Finally, Section 6.4 presents an ablation of two components of our approach.

## 6.1 EXPERIMENTAL SETUP

**Datasets.** Our evaluation utilizes six challenging models from CrossSDF Walker et al. (2025). Cross-sections of CT scans are treated as input point clouds. We follow the CrossSDF protocol in using both aligned and non-aligned cross-section orientations. All models consist of a single

|  | Method | Alveolis | | Cerebral | | Coronaries | | Coro | | Heart | | Pulmonary | |
|---|---|---|---|---|---|---|---|---|---|---|---|---|---|
|  |  | Align | NonAlign | Align | NonAlign | Align | NonAlign | Align | NonAlign | Align | NonAlign | Align | NonAlign |
| CD | CrossSDF | 0.35 | 0.56 | 0.24 | 0.26 | 0.28 | 0.46 | 0.21 | 0.29 | 0.38 | 0.82 | 0.24 | 0.34 |
|  | ORex | 3.6 | 4.8 | 12 | 2 | 10 | 1.6 | 0.48 | 0.33 | 4.6 | 8.4 | 12 | 5.6 |
|  | ScreenedPoisson | 0.44 | 0.57 | 0.39 | 0.44 | 0.46 | 0.67 | 1.3 | 0.95 | 1.0 | 0.85 | 0.52 | 0.59 |
|  | POCO | 4.1 | 4.5 | 0.36 | 0.79 | 1.1 | 0.72 | 1.1 | 3.4 | 2.7 | 4.2 | 2.4 | 2.0 |
|  | Neural-IMLS | 13 | 12 | 14 | 17 | 19 | 17 | 9 | 10 | 19 | 18 | 15 | 15 |
|  | Ours | 0.41 | 0.53 | 0.21 | 0.22 | 0.26 | 0.33 | 0.16 | 0.18 | 0.32 | 0.66 | 0.21 | 0.28 |
| HD | CrossSDF | 14 | 6.1 | 3.5 | 4 | 5.8 | 4.7 | 4.5 | 2.8 | 11 | 10 | 10 | 7.3 |
|  | ORex | 21 | 35 | 8.1 | 19 | 18 | 27 | 9 | 11 | 26 | 42 | 36 | 21 |
|  | ScreenedPoisson | 17 | 8.1 | 16 | 25 | 6.1 | 5.4 | 10 | 7.5 | 26 | 9.2 | 10 | 7.7 |
|  | POCO | 32 | 29 | 15 | 41 | 7.1 | 9.2 | 8.8 | 18 | 24 | 25 | 20 | 16.4 |
|  | Neural-IMLS | 68 | 51 | 33 | 48 | 55 | 67 | 37 | 39 | 59 | 61 | 52 | 41 |
|  | Ours | 11.8 | 8.6 | 2.8 | 2.4 | 4.6 | 5.2 | 5.3 | 4.5 | 6.2 | 7.8 | 8.2 | 9.3 |
| CC | GT | 1 | 1 | 1 | 1 | 3 | 3 | 1 | 1 | 1 | 1 | 1 | 1 |
|  | CrossSDF | 6 | 33 | 2 | 5 | 5 | 35 | 2 | 4 | 68 | 176 | 2 | 3 |
|  | OReX | 939 | 2119 | 40 | 421 | 219 | 94 | 13 | 4 | 223 | 511 | 914 | 261 |
|  | ScreenedPoisson | 53 | 1187 | 3 | 23 | 81 | 160 | 9 | 62 | 923 | 1006 | 6 | 17 |
|  | POCO | 67 | 91 | 2 | 78 | 14 | 46 | 4 | 279 | 70 | 134 | 21 | 134 |
|  | Neural-IMLS | 1 | 1 | 1 | 1 | 1 | 1 | 1 | 1 | 1 | 1 | NA | NA |
|  | Ours | 1 | 1 | 1 | 1 | 3 | 3 | 1 | 1 | 1 | 1 | 1 | 1 |

Table 1: Quantitative results on thin structures across different methods and metrics. The table compares the Chamfer Distance(CD)×100, Hausdorff Distance(HD)×100, and number of Connected Components(CC), under both aligned and non-aligned versions of each structure. Note on the Coronaries dataset: because our core algorithm is designed to reconstruct a single simply connected component, we leveraged prior knowledge to pre-segment the Coronaries point cloud into three anatomical components. Each component is reconstructed independently, and the results were aggregated.

connected component, except for the Coronaries model, which has three distinct segments. These segments are separated using single-linkage clustering and processed individually. For synthetic data, we add varying levels of noise and outliers to the Stanford Bunny model to compare our method against baselines.

**Evaluation Protocol.** Following CrossSDF, we assess reconstruction quality using three metrics. Chamfer Distance (CD) and Hausdorff Distance (HD) are reported to measure geometric fidelity against the ground truth mesh. For thin structures, we also report the number of Connected Components (CC) to verify topological consistency and structural integrity.

**Implementation.** We implemented a high-performance solver for the sparse linear system in Equation (16) on tetrahedral meshes. Our approach is a direct application of algorithm-system co-design: we fused the conjugate gradient (CG) method and its geometric multigrid preconditioner into a single persistent GPU kernel. This design contains all CG iterations and multigrid propagations Bolitho et al. (2007), eliminating costly memory round-trips. By leveraging CUTLASS Thakkar et al. (2025), we further optimized sparse matrix arithmetic and memory access patterns for the NVIDIA GH200 GPU. As a result, a single iteration of our solver for Equation (16) on a 100k point cloud completes in 0.13 milliseconds.

## 6.2 Analysis of 3D Medical Reconstruction

We compare our method against five baselines: CrossSDF Walker et al. (2025) and OReX Sawdayee et al. (2023), which are tailored for reconstruction from cross-sections, and three state-of-the-art general-purpose methods, ScreenedPoisson Kazhdan & Hoppe (2013), POCO Boulch & Marlet (2022), and Neural-IMLS Wang et al. (2024).

Quantitative results are presented in Table 1. Our approach consistently outperforms all baselines across all three metrics (CD, HD, CC). Notably, it is the only method to correctly reconstruct the ground truth number of connected components. While Neural-IMLS is the second-best in topological accuracy, our method achieves significantly better geometric fidelity. Compared to CrossSDF, the closest competitor in geometric terms, our approach reduces CD and HD by 15.8% and 9.62%, respectively. In contrast, CrossSDF produces a large number of disconnected components for complex structures like the Alveoli and Heart.

Qualitative comparisons in Figure 3 visually confirm these findings. Our reconstructions are the only ones that simultaneously achieve high geometric detail comparable to the ground truth while

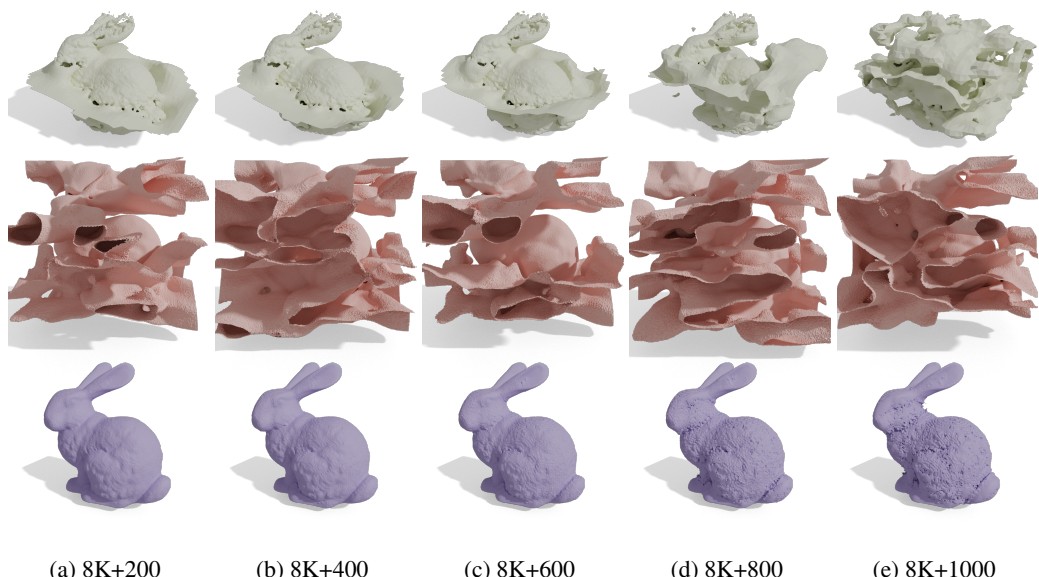

|  | (a) 8K+200 | (b) 8K+400 | (c) 8K+600 | (d) 8K+800 | (e) 8K+1000 |

Figure 4: Reconstruction of a synthetic shape at various noise. Top row: Screened PSR; Middle row: SALD; Bottom row: Ours. Each column corresponds to a different number of outliers.

| | Method | 8K+200 | 8K+400 | 8K+800 | 4K+200 | 4K+400 | 4K+800 | 2K+200 | 2K+400 | 2K+800 | 1K+200 | 1K+400 | 1K+800 |
|---|---|---|---|---|---|---|---|---|---|---|---|---|---|
| CD | ScreenedPoisson | 0.35 | 0.56 | 0.24 | 0.26 | 0.28 | 0.46 | 0.21 | 0.29 | 0.38 | 0.82 | 0.24 | 0.34 |
| | SALD | 3.6 | 4.8 | 12 | 2 | 10 | 1.6 | 0.48 | 0.33 | 4.6 | 8.4 | 12 | 5.6 |
| | Ours | 0.18 | 0.26 | 0.33 | 0.28 | 0.31 | 0.36 | 0.24 | 0.26 | 0.42 | 0.81 | 0.93 | 1.1 |
| HD | ScreenedPoisson | 14 | 6.1 | 3.5 | 4 | 5.8 | 4.7 | 4.5 | 2.8 | 11 | 10 | 10 | 7.3 |
| | SALD | 21 | 35 | 8.1 | 19 | 18 | 27 | 9 | 11 | 26 | 42 | 36 | 21 |
| | Ours | 12.1 | 9.6 | 2.8 | 3.2 | 4.3 | 3.8 | 3.1 | 3.5 | 8.9 | 7.6 | 12.6 | 13.2 |
| CC | ScreenedPoisson | 6 | 33 | 2 | 5 | 5 | 35 | 2 | 4 | 68 | 176 | 2 | 3 |
| | SALD | 939 | 2119 | 40 | 421 | 219 | 94 | 13 | 4 | 223 | 511 | 914 | 261 |
| | Ours | 1 | 1 | 1 | 1 | 1 | 1 | 1 | 1 | 1 | 1 | 1 | 1 |

Table 2: The evaluation protocol is the same as Table 1. Each dataset is denoted as #inliers + #outliers.

preserving the correct topology. This demonstrates the superiority of our method in maintaining both geometric and topological integrity.

## 6.3 ANALYSIS OF 3D SYNTHETIC SHAPE RECONSTRUCTION

We compare our approach with two general-purpose 3D reconstruction approaches, namely ScreenedPoisson and SALD Atzmon & Lipman (2021). For each clean model, we generate an input by varying two parameters: the number of points on the original shape and the number of outliers perturbed from surface points along their normals through a random offset between $[-b, b]$, where $b$ is set as $0.25$ of the length of the bounding-box diagonal.

Table 2 shows the quantitative performance of our approach and the baseline approaches. We can see that our approach significantly outperforms baseline approaches in terms of robustness against outliers. In the presence of high outlier ratios, our approach still achieves correct topology and high geometric fidelity. This is attributed to the robust optimization module of our approach for removing outliers. In contrast, both SALD and ScreenedPoisson require clean inputs for successful reconstructions. Figure 4 shows qualitative results that are consistent with quantitative results.

## 6.4 ABLATION STUDY

We conduct an ablation study in 2D to demonstrate the importance of each component of our framework.

**No initial orientation.** Without initial orientation, our reconstruction fails completely, as shown in Figure 5. Inconsistent orientations lead to a poor initial surface, causing essential inlier edges to be incorrectly pruned. Recovering from this state is difficult, as topological priors alone do not enforce geometric correctness.

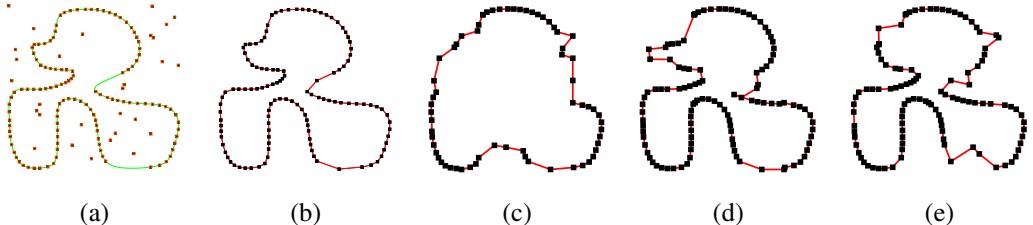

Figure 5: (a) Input. (b) Full approach. (c) No initial orientation. (d) No robust optimization. (e) No alternating optimization.

**No robust optimization.** The $L^0$ norm in the optimization of the correction surface $\Lambda$ is critical for promoting sparsity and smoothness in the reconstruction. Replacing it with a less robust regularizer, such as the $L^1$ norm, can result in non-smooth artifacts.

**No alternating optimization.** Alternating optimization is essential for handling significant numbers of outliers. A single optimization pass is insufficient to identify all outliers. Our iterative approach allows the correction surface $\Lambda$ to progressively refine the inlier set $\mathcal{F}_0$ across iterations, gradually removing all outliers and converging to the correct reconstruction.

## 7 CONCLUSIONS, DISCUSSIONS, AND FUTURE WORK

In this paper, we introduce an approach that guarantees the reconstruction of a closed, simply connected mesh from a noisy point cloud. Our method generalizes the winding number field of curves Feng et al. (2023) to three dimensions: it takes as input a collection of oriented triangles from a 3D triangulation and produces a closed, simply connected mesh. We develop strategies to infer consistent triangle orientations from a tetrahedral mesh and to remove input outliers through an alternating optimization scheme. Experimental results on both real and synthetic benchmark datasets demonstrate the effectiveness of our approach.

Our approach assumes that input inliers are accurate, as they are vertices of the reconstructed mesh. This assumption is reasonable because most sensor noise in LiDAR or CT scans is white or salt-and-pepper noise. In the presence of Gaussian noise, however, a better strategy is to place vertices along the edges of the underlying 3D tetrahedral mesh, similar to the MarchingCube approach Lorensen & Cline (1987) in Poisson surface reconstruction Kazhdan et al. (2006). In future work, we plan to develop strategies to smooth the final function $\mathbf{w}$ in Equation (11), which can be used with MarchingTetrahedron DOI & KOIDE (1991) for surface reconstruction.

There are ample opportunities for future research. While this paper has focused on closed, simply connected shapes, we plan to extend our framework to handle arbitrary topologies by developing tools to control the harmonic component $\boldsymbol{\gamma}$ in the HHD. Another future direction is to leverage this representation to learn generative models of 3D shapes with controllable topologies.

### ACKNOWLEDGMENTS

This work was supported by NSF-2047677, NSF-2413161, NSF-2504906, NSF-2515626, and GIFTs from Adobe and Google. This work was supported by computing support on the Vista GPU Cluster through the Center for Generative AI (CGAI) and the Texas Advanced Computing Center (TACC) at UT Austin.

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

## A   DISCRETE EXTERIOR CALCULUS AND HODGE OPERATORS

We briefly summarize the discrete exterior calculus (DEC) operators used throughout the paper. In DEC, a discrete $k$-form is a quantity assigned to oriented $k$-dimensional mesh elements that can be integrated over them. On a tetrahedral mesh, discrete differential forms are associated with mesh elements: 0-forms with vertices, 1-forms with edges, and 2-forms with faces. These operators provide discrete analogues of the gradient, curl, and divergence from vector calculus.

**Exterior derivative and codifferential.**   The exterior derivative operators $d_k : \Omega^k \to \Omega^{k+1}$ map $k$-forms to $(k+1)$-forms. In particular, $d_0$ corresponds to the discrete gradient and $d_1$ to the discrete curl. The codifferential operators $\delta_k : \Omega^k \to \Omega^{k-1}$ are adjoints of $d_k$. Here, $\delta_1$ corresponds to divergence, while $\delta_2$ is the adjoint of the curl operator.

**Definition A.1 (Discrete Curl and Divergence)** *Let $\boldsymbol{\omega}$ be a discrete 1-form defined on oriented edges.*

- *Discrete curl $d_1$ maps edge-based values to faces and measures the circulation of $\boldsymbol{\omega}$ around each face. For an oriented face $f = (i, j, k)$,*

$$(d_1\boldsymbol{\omega})_f = \omega_{ij} + \omega_{jk} + \omega_{ki}.$$

- *Discrete divergence $\delta_1$ maps edge-based values to vertices and measures the net flow leaving each vertex:*

$$(\delta_1\boldsymbol{\omega})_i = \sum_{j\in\mathrm{adj}(i)} w_{ij}\, \omega_{ij},$$

*where $\mathrm{adj}(i)$ denotes the vertices adjacent to $i$ and $w_{ij}$ are geometry-dependent edge weights.*

*In DEC terminology, $d_1$ is the exterior derivative and $\delta_1$ is its adjoint.*

**Definition A.2 (Discrete Hodge Laplacians)** *The scalar potentials in the Hodge–Helmholtz decomposition are obtained by solving Poisson equations involving discrete Hodge Laplacians:*

- *The 0-form Laplacian*

$$\Delta_0 = \delta_1 d_0$$

*operates on vertex-based scalar functions.*

- *The 2-form Laplacian*

$$\Delta_2 = d_1 \delta_2$$

*operates on face-based quantities.*

*These operators are sparse and can be assembled efficiently.*

## B   HODGE-HELMHOLTZ DECOMPOSITION

This section presents a formal introduction to the Hodge-Helmholtz decomposition of the discrete 1-form $\boldsymbol{\omega} = Du$, where $D$ denotes the discrete gradient operator mapping the vertex-based scalar field $u$ to a 1-form defined on edges.

The Hodge-Helmholtz decomposition states that any discrete 1-form $\boldsymbol{\omega}$ can be uniquely and orthogonally decomposed into three components: an exact part, a coexact part, and a harmonic part:

$$\boldsymbol{\omega} = d_0\boldsymbol{\alpha} + \delta_2\boldsymbol{\beta} + \boldsymbol{\gamma}. \tag{17}$$

Here, $\boldsymbol{\alpha}$ is a discrete 0-form (a scalar potential on vertices) and $\boldsymbol{\beta}$ is a discrete 2-form (a scalar potential on faces). The component $d_0\boldsymbol{\alpha}$ is curl-free, $\delta_2\boldsymbol{\beta}$ is divergence-free, and the harmonic component $\boldsymbol{\gamma}$ is both curl-free and divergence-free. This harmonic part represents the de Rham cohomology class of $\boldsymbol{\omega}$. By de Rham's theorem, this class is isomorphic to the nontrivial 1-homology of

the domain. In our context, $\gamma$ provides the dual representation of all non-bounding cycles implicitly induced by the topological errors in the input.

To compute the Hodge–Helmholtz decomposition, we first compute the potentials $\alpha$ and $\beta$ by solving two decoupled Poisson systems, as described in Appendix B.1 and Appendix B.2, respectively. Then the harmonic component $\gamma$ is obtained as the residual.

### B.1    COMPUTING THE EXACT (CURL-FREE) COMPONENT

To find the exact (curl-free) component $d_0\alpha$, we first solve for the scalar potential $\alpha$. Our strategy is to eliminate the other components of the Hodge-Helmholtz decomposition by applying the discrete divergence operator $\delta_1$ to Equation (17):

$$\delta_1\boldsymbol{\omega} = \delta_1 d_0\boldsymbol{\alpha} + \delta_1\delta_2\boldsymbol{\beta} + \delta_1\boldsymbol{\gamma}.$$

The last two terms vanish: $\delta_1\delta_2 = 0$ because the divergence of a curl-like field is zero and $\delta_1\boldsymbol{\gamma} = 0$ because harmonic components are divergence-free. Therefore,

$$\delta_1\boldsymbol{\omega} = \delta_1 d_0\boldsymbol{\alpha}. \tag{18}$$

Since $\delta_1 d_0$ is the discrete 0-form Laplacian $\Delta_0$, we obtain the Poisson equation

$$\Delta_0\boldsymbol{\alpha} = \delta_1\boldsymbol{\omega}.$$

In matrix form, this becomes

$$L_0\boldsymbol{\alpha} = \mathbf{b}_\alpha,$$

where $L_0$ is the dihedral-angle Laplacian, $\boldsymbol{\alpha}$ is the vector of unknown vertex potentials, and $\mathbf{b}_\alpha = \delta_1\boldsymbol{\omega}$ is computed using the discrete divergence operator.

This sparse linear system is solved subject to a constraint ensuring uniqueness (e.g., fixing one vertex value or enforcing a zero-mean solution). Once $\boldsymbol{\alpha}$ is obtained, the exact component is recovered as $d_0\boldsymbol{\alpha}$.

### B.2    COMPUTING THE CO-EXACT (DIVERGENCE-FREE) COMPONENT

To extract the co-exact (divergence-free) component $\delta_2\boldsymbol{\beta}$, we solve for the 2-form potential $\boldsymbol{\beta}$. Analogous to the previous subsection, we eliminate the other components of the decomposition by applying the discrete curl operator $d_1$ to Equation (17):

$$d_1\boldsymbol{\omega} = d_1 d_0\boldsymbol{\alpha} + d_1\delta_2\boldsymbol{\beta} + d_1\boldsymbol{\gamma}.$$

The first and last terms vanish: $d_1 d_0 = 0$ because the curl of a gradient is zero, and $d_1\boldsymbol{\gamma} = 0$ because harmonic components are curl-free. Thus,

$$d_1\boldsymbol{\omega} = d_1\delta_2\boldsymbol{\beta}.$$

Assuming the co-closed component of $\boldsymbol{\beta}$ is zero, $d_1\delta_2$ reduces to the discrete 2-form Laplacian $\Delta_2$, yielding the Poisson equation

$$\Delta_2\boldsymbol{\beta} = d_1\boldsymbol{\omega}.$$

In matrix form, this becomes

$$L_2\boldsymbol{\beta} = \mathbf{b}_\beta,$$

where $L_2$ is the matrix representation of $\Delta_2 = d_1\delta_2$, $\boldsymbol{\beta}$ is the vector of unknown face-based potentials, and $\mathbf{b}_\beta = d_1\boldsymbol{\omega}$ is computed using the discrete curl operator.

This sparse linear system is solved for $\boldsymbol{\beta}$, after which the co-exact component is recovered as $\delta_2\boldsymbol{\beta}$.

### B.3    HOMOLOGICAL GUARANTEES VIA HELMHOLTZ-HODGE DECOMPOSITION

De Rham's theorem states that for a smooth manifold, its $k$-th de Rham cohomology group is isomorphic to its singular $k$-th homology group with real coefficients De Rham (1931). In our discrete setting, this means that the space of harmonic $k$-forms is isomorphic to the space of non-trivial $k$-cycles. For our 3D problem, the harmonic 1-form $\gamma$, obtained from the HHD of the field $Du$,

provides a dual representation of the nonbounding 2-cycles in $\Gamma$. A 2-cycle is nonbounding if it does not arise as the boundary of any 3-chain, i.e., if it represents a topological hole or handle. By algorithmically isolating and removing the influence of $\boldsymbol{\gamma}$, we construct a final jump surface that is guaranteed to bound a volume and hence homologically trivial, i.e., a disjoint union of spheres. In $\mathbb{R}^3$, every sphere is simply connected.

## C  VOLUMETRIC LAPLACIAN

This section details the definition of $w_{ij}^T$, where $(i, j)$ is an edge in a tetrahedron $T$. As shown in Figure 6, let the four vertices of $T$ be $\boldsymbol{v}_i$, $\boldsymbol{v}_j$, $\boldsymbol{v}_k$, and $\boldsymbol{v}_l$. Define $\alpha_{ij}$ as the angle between edges $\boldsymbol{v}_j\boldsymbol{v}_k$ and $\boldsymbol{v}_i\boldsymbol{v}_k$, and $\beta_{ij}$ as the angle between edges $\boldsymbol{v}_j\boldsymbol{v}_l$ and $\boldsymbol{v}_i\boldsymbol{v}_l$. Let $\theta_{ij}^{kl}$ denote the dihedral angle between faces $\boldsymbol{v}_i\boldsymbol{v}_j\boldsymbol{v}_k$ and $\boldsymbol{v}_i\boldsymbol{v}_j\boldsymbol{v}_l$. Following Liao (2025), the edge weight is defined as

$$w_{ij}^T = \frac{\|\boldsymbol{v}_i - \boldsymbol{v}_j\|}{8}\Big(2\cot\alpha_{ij}\cot\beta_{ij} - \cos\theta_{ij}^{kl}\big(\cot^2\alpha_{ij} + \cot^2\beta_{ij}\big)\Big).$$

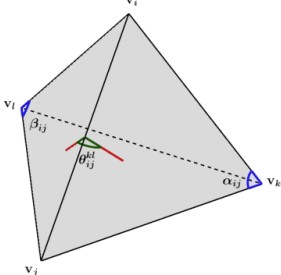

Figure 6: Edge weight $w_{ij}^T$

