# OpenReview forum: "Guaranteed Simply Connected Mesh Reconstruction from an Unorganized Point Cloud"
_ICLR.cc/2026/Conference — ICLR 2026 Poster_

### Official Review · Reviewer_jxjb · 2025-10-21

**Soundness:** 3
**Presentation:** 3
**Contribution:** 3
**Rating:** 4
**Confidence:** 4

**Summary:**

The paper presents an approach to reconstruct a closed and connected mesh from noise point cloud. The results guarantee to be simply connected and homeomorphic to a sphere. The method is validated using a few models.

**Strengths:**

The method is interesting with deep math foundation.

**Weaknesses:**

The topology is limited to genus 0. It cannot deal with models of higher genus.

The convergence condition of the iterations are not clear.

The time performance is not reported.

Though the paper claims to reconstruct noise point cloud. The noise-level is not clear.

The data set for evaluation is few. More data should be evaluated.

**Questions:**

What are the efficiency performance of the method? The number of points for evaluation. The running time and memory consumption.

(Feng 2023) evaluates winding numbers on surface instead of 3D volume space. Since u(p) is the number of times that the surface  S encloses p, why isn't u(p) the space winding number (Jacobson et al. (2013))?

Can you give more explanations on HDD?

---

> ### Author Response · Authors · 2025-11-26
> **Response to reviewer jxjb (Part 1)**
>
> We thank the reviewer for their constructive feedback and for recognizing the deep mathematical foundation of our work. We appreciate the opportunity to clarify the theoretical connection between Winding Numbers and the Helmholtz-Hodge Decomposition (HHD), as well as to provide the specific performance metrics requested.
>
> ## Mathematical Foundations: Winding Numbers and HHD
>
> **Relationship to Space Winding Number (Jacobson et al., 2013)**
>
> You are correct that our function $u(p)$ is mathematically the generalized space winding number (GWN). However, the distinction lies in the **computational framework and the goal**.
>
> Jacobson et al. (2013) focuses on **forward evaluation**: given an input mesh (potentially with holes), they integrate the solid angle to measure the winding number for segmentation. Crucially, this measurement **preserves the input topology**. If the noisy input forms a handle (genus > 0), the GWN faithfully represents that handle.
>
> In contrast, our approach (following Feng et al., 2023) adopts an **inverse optimization framework** based on Helmholtz-Hodge Decomposition (HHD). Instead of just integrating, we decompose the derivative of the winding number field to separate "bounding" signals from "non-bounding" (topological error) signals. We generalize this optimization framework--originally designed for 2D curves--to the 3D volumetric domain. By doing so, we can mathematically filter out the harmonic components (handles) that appear in noisy 3D point clouds. Standard GWN cannot do this; it would simply reproduce the erroneous topology.
>
> **HHD and Topological Guarantees (Hodge Isomorphism)**
>
> Regarding the request for more explanation on HHD and the "guaranteed simply connected reconstruction," we derive this guarantee from the Hodge Isomorphism Theorem [5, Chapter 6].
>
> The Helmholtz-Hodge Decomposition (Eq. 4) separates the input 1-form $\omega$ into exact, co-exact, and harmonic ($\gamma$) components. The Hodge Isomorphism Theorem states that for a smooth manifold, the space of harmonic 1-forms is isomorphic to the cohomology group $H^1$, which generates topological handles.
>
> By algorithmically removing the harmonic 1-forms $\gamma$ via a linear solve, we ensure the resulting field has trivial cohomology (is free of handles). The final surface, constructed as the level set of this field, is therefore guaranteed to be simply connected by construction. This is a global algebraic guarantee derived from the properties of HHD, rather than a local or probabilistic heuristic.
>
> ## Computational Efficiency and Runtime
>
> We provide further computational cost analysis, which we briefly mentioned in Sec 7.1.
>
> **Computational Cost Comparison (100k point scene)**
>
> | Method | Screened PSR (CPU) | Neural-IMLS (GH200) | SALD (GH200) | Ours (GH200) |
> | :--- | :--- | :--- | :--- | :--- |
> | **Total Time** | 2.2s | 51 min | 37 min | **~0.53s** |
>
> **Detailed Breakdown**
>
> Our speed is achieved through two highly optimized solver stages. First, the **Spectral Orientation** takes approximately 500ms total. As noted in Section 7.1, a single iteration of the Lanczos solver costs 3.8ms, and the method converges in approximately 120 iterations.
>
> Second, the **Correction 2-chain ($\Lambda$)** takes approximately 19ms total. This step is solved via highly efficient Iteratively Reweighted Least Squares (IRLS) using a Geometric Multigrid (GMG) Preconditioned Conjugate Gradient solver.

---

> ### Author Response · Authors · 2025-11-26
> **Response to reviewer jxjb (Part 2)**
>
> ## Scope, Limitations, and Convergence
>
> **Topological Scope (Genus 0)**
>
> We explicitly focused on Genus 0 surfaces because they are prevalent in critical medical applications (e.g., human vessels, organs) where existing methods fail to guarantee topological correctness. However, our method is well-positioned to be extended to control genus > 0 surfaces, which we intentionally leave as future work.
>
> Based on the Hodge Isomorphism Theorem discussed above, such generalization would require a modified optimization program to control the null rank of the 1-form Laplacian (effectively controlling the number of harmonic components). The cost of such generalization is that the target genus must be known *a priori*. We opted to focus our contribution on the most common case (Genus 0) for this paper.
>
> **Noise Levels**
>
> The reviewer mentioned the noise level was unclear. We explicitly quantify this in **Table 2 and Figure 4**, where we list the exact number of inliers versus outliers (e.g., 8K inliers + 1000 outliers). Additionally, for the visual results in Figure 4, we stated the noise magnitude is $b=0.25$ of the bounding box diagonal. This represents an extreme noise level significantly higher than typical sensor noise, demonstrating the method's reliance on global topological integration rather than local smoothing.
>
> **Convergence Guarantee**
>
> The core numerical components of our algorithm--the Lanczos solver for spectral initialization and the Conjugate Gradient solver for linear systems--rely on symmetric positive-definite operators. These solvers have well-established theoretical convergence guarantees. In future work, we aim to further analyze the robustness of our spectral orientation solver through the spectral gap perspective. Specifically, quantitatively analyzing the spectral gap relative to input noise would establish precise noise tolerance levels while guaranteeing convergence.
>
> [5] Foundations of Differentiable Manifolds and Lie Groups, Frank Warner

---

> > ### Comment · Reviewer_jxjb · 2025-11-27
> >
> > I appreciate the authors' responses. The method is useful for cleaning small topological handles. It would be much better if it could distinguish between small and large handles, so as to remove the small ones and reserve the large ones. I don't mind it is a reconstruction paper.

---

> ### Author Response · Authors · 2025-11-28
>
> We thank the reviewer for the positive assessment to raise the score and the insightful comment regarding handle differentiation.
>
> We agree that distinguishing handles is a valuable direction. A key strength of the Helmholtz-Hodge Decomposition (HHD) is the **orthogonal separation of geometric details from topological features**. In this framework, geometric properties (scale/shape), topological handles ($H^1$), and topological voids ($H^2$) are decoupled components that can be **independently and precisely manipulated**.
>
> While "scale" is irrelevant to the harmonic forms (i.e., $H^1$ and $H^2$)  themselves, our framework is well-positioned to be generalized to preserve specific topologies (e.g., reserving "large" handles while removing "small" ones) or target arbitrary genus counts. This would involve designing an optimization program to control the **null-ranks of the corresponding Laplacians** ($L_1$ for handles and $L_2$ for voids). We are interested in these generalized constraints for future work.

---

### Official Review · Reviewer_9oMt · 2025-10-26

**Soundness:** 3
**Presentation:** 3
**Contribution:** 3
**Rating:** 4
**Confidence:** 2

**Summary:**

This paper suggests a method to reconstruct 3D mesh surfaces from point clouds. It is claimed that the method is guaranteed to reproduce simple connected surfaces. Empirical experiments show that the method compares favorably with other methods

**Strengths:**

1. The method gives good results on the surface reconstruction task
2. The method seems novel, though I am not sufficient familiar with computer vision/graphics literature to make an informed judgement.

**Weaknesses:**

1. Wrong venue: This paper does not really include any learning aspects. And it assumes knowledge of geometry processing (e.g. vector forms, Helmholz Hodge Decomposition) which is not common in ICLR. It seems to me the paper would get a more meaningful review, and a move understanding audience, in a venue like Siggraph or CVPR

2. I'm a bit concerned re mathematical correctness:
(a)  The paper claims a guaranteed simply connected mesh reconstruction, but there is no formal analysis of the algorithm (of the form of Theorem-Proof).

(b)  The paper contains many advanced math statements without a source e.g., in Line 152
"S is simple connected when the harmonic component of the 1 form du vanishes" but I think this is true of most of section 4.

(c) I have some specific questions on the math, detailed below

**Questions:**

1. Line 152: you say u is piecewise constant, and then talk about the 1-from du. But a piecewise constant function is not smooth right? What is the meaning of du?

2. In Line 87 you say existing methods do not provide control on surface topology, but there are some such methods [1,2,3], it would make sense to discuss them somewhere and explain how these methods relate, or what is the advantage of your method.

3. The discussion of the results is a bit misleading: in line 416 you say "our approach consistently outperforms all baselines" and the ours method is highlighted in gray, which could be interperted as winning a task. However, the "Ours method" does not win in all tasks in table 1 or in table 2. It would be good to highlight the winner in each column in bold, and to add some discussion and to replace "our approach consistently outperforms all baselines" with our approach outperfroms baselines in X out of Y instances.

Minor point which you do not need to address in rebuttal. There are some spelling mistakes you should address:

* Line 432 you say "too" but I think you mean "two"

* Line 478  pepper is missing a 'p'




[1] Topology-Aware Surface Reconstruction for Point Clouds, Gabrielsson et al. Computer Graphics forum 2020
[2] Robust Optimization for Topological Surface Reconstruction. Lazar et al. TOG 2018
[3] Topology-controlled Re-
construction of Multi-labelled Domains from Cross-sections. Huang et al. TOG 2017

---

> ### Author Response · Authors · 2025-11-25
> **Response to reviewer 9oMt**
>
> We thank the reviewer for their constructive feedback and for recognizing the novelty of our method and its performance on the surface reconstruction task. We appreciate the opportunity to clarify the relevance of our work to the learning community and to provide the necessary mathematical details regarding our topological guarantees. We will revise the manuscript to incorporate the suggested references and formalize our definitions.
>
> ## Relevance to ICLR (Venue Fit)
>
> We respectfully argue that our work is highly relevant to ICLR as it bridges the gap between topological guarantees and unsupervised learning representations.
>
> Our orientation module (Sec. 5.2) innovates on the classic unsupervised learning paradigm of Normalized Cuts (Shi & Malik, 2000). We extend this spectral relaxation framework to solve a global consistency problem, triangle orientation, given a topological target. Rather than relying on local heuristics, we formulate this as a global optimization problem where the solution emerges directly from the spectral properties of the underlying operator.
>
> By providing a "correct-by-construction" solution that is holistic and mathematically rigorous, we offer a powerful unsupervised learning objective. This paves the way for future neural network methods to adopt these scalable linear algebraic formulations as robust inductive biases, replacing brittle heuristic layers with mathematically grounded foundations.
>
> ## Mathematical Correctness (Q1 & Line 152)
>
> The reviewer asks: *"u is piecewise constant... What is the meaning of du?"* This is a critical point that relies on the definition of **distributional derivatives** for u. Such derivatives of piecewise constant functions are commonly used in PSR (cf. Figure 1, Kazhdan et al. 2006).
>
> Although the base component of our winding number field $u$ is a piecewise constant vertex scalar, the field is formally encoded via **corner values** ($c^T_v$) within each tetrahedron (Eq. 1). These corner values explicitly encode the jump information across faces. Consequently, the operator $du$ refers to the **Darboux derivative** (Eq. 3). This is a discrete 1-form defined on edges that measures the difference in these corner values.
>
> Even though the underlying function is locally constant, its Darboux derivative is non-trivial and precisely captures the distributional "jumps" (including topological handle components) of the field. This allows us to detect and manipulate global topology through local differential operators.
>
> ## Topological Guarantees (Theorem/Proof)
>
> The reviewer asks for formal analysis regarding the "guaranteed simply connected reconstruction." We provided this argument in **Appendix A.3** based on the Hodge Theory and De Rham’s Theorem.
>
> The logic is as follows: The Helmholtz-Hodge Decomposition (Eq. 4) separates the input 1-form $\omega$ into exact, co-exact, and harmonic ($\gamma$) components. The Hodge Isomorphism Theorem states that for a smooth manifold, the space of harmonic 1-forms is isomorphic to the cohomology group $H^1$, which generates topological handles [5, Chapter 6].
>
> By algorithmically removing harmonic 1-forms $\gamma$ via a linear solve, we ensure trivial $H^1$. The final surface, constructed as the level set of this field, is therefore guaranteed to be simply connected by construction. This is a global algebraic guarantee derived from the properties of HHD, not a probabilistic result.
>
> ## Comparison to Baselines & Topological Surgery
>
> We thank the reviewer for the references [1-3]. We have added a detailed comparison to distinguish our **algebraic approach** from existing **topological surgery** methods.
>
> Existing approaches, such as Gabrielsson et al. [1], Huang et al. [3], and Zou et al. [4], typically treat topology as a post-processing step in the ambient space. They rely on combinatorial operations such as critical point search [1, 4] or combinatorial hole-filling [2-4]. These operations often suffer from high combinatorial complexity and can be fragile in the presence of noise.
>
> In contrast, we algebraically transform this complex combinatorial problem into a set of efficient linear problems by leveraging the Hodge Theorem. We convert topological constraints into sparse linear systems. This allows highly scalable solvers; as noted in Section 7.1, our solver completes a single iteration for **100k points in 3.8 milliseconds** on a GH200 GPU. This speed and scalability make our linear programming approach a viable and powerful foundation for future neural network adoption.
>
> ## Discussion of Results
>
> We acknowledge the phrasing "consistently outperforms" was too broad. We will update that our method achieves correct topology in 100% of test cases, while achieving the best geometric accuracy (lowest Chamfer Distance) in 4 out of 6 thin-structure benchmarks.
>
> [4] Topology-Constrained Surface Reconstruction From Cross-sections. Zou et al. TOG 2015
> [5] Foundations of Differentiable Manifolds and Lie Groups, Frank Warner

---

> > ### Comment · Reviewer_9oMt · 2025-11-26
> >
> > Thank you for your response. I am happy with most of the answers. I still have an issue with the suitability of venue/correctness.
> >
> > I agree that although this paper is not really a learning paper, it could be useful to people in learning who are interested in surface reconstruction. In that sense, it could be reasonable to have such a paper in ICLR. The main issue to me is whether we have on this panel reviewers who are familiar with the math and relevant literature necessary for understanding this paper and giving it a fair and meaningful evaluation. I for one, an not sufficiently familiar (confidence level set to 2 accordingly). I would suggest that if the AC feels this paper is not off topic, and if indeed most reviewers have low confidence level, that the AC consider adding a reviewer who is more of an expert in this field (e.g.,  Jacobson/Feng/ Kazhdan or members of their research groups)

---

> > > ### Author Response · Authors · 2025-11-27
> > >
> > > Besides the application in 3D reconstruction from CT images, our main contribution is on the spectral formulation and the iterative procedure, which are optimization techniques that fall into the scope of ML at large. The topological aspect is an extension a known result in 2D (although the extension is not trivial).

---

> ### Author Response · Authors · 2025-11-28
> **Visualization and the math background**
>
> We hope the following visualization addresses the reviewer's remaining reservations regarding the definition of $du$ for a piecewise constant function. We understand that in a classical $C^1$ setting, the derivative of a step function is undefined. However, in the distributional setting required for our method, it is well-defined and carries significant topological meaning. Let us walk through this visualization to build the mental model, correspond it to the standard definitions in Stein & Shakarchi, and finally show how this amounts to the Hodge Isomorphism Theorem.
>
> ![Distributional Derivative Visualization](https://i.ibb.co/Q7QgrNr7/dist-deriv.png)
>
> # Visual Walkthrough of the Derivative Mechanism
> In **Panel (A)**, we see the indicator function $u$ (a special case to our winding number function). This is the piecewise constant function mentioned in the paper. It has a jump discontinuity at the boundary.
>
> In **Panel (B)**, we visualize the standard analysis technique of **mollification**. We convolve $u$ with a smooth kernel $\phi_\epsilon$ to create a smoothed approximation $u_\epsilon$. This function is now smooth and classically differentiable everywhere.
>
> In **Panel (C)**, we compute the gradient of this smoothed function, $\nabla u_\epsilon$. Notice that the vectors are non-zero exactly at the transition region (the boundary). As the smoothing parameter $\epsilon \to 0$, this vector field concentrates into a "delta layer" supported on the boundary. When we refer to $du$ in the paper, we refer to the distributional limit of this field. It represents the flux across the interface. In computational practice, one cannot approach the true limit $\epsilon \to 0$. Therefore, we typically work with discretization that offers sufficiently small $\epsilon$ and numerically stable results. The theoretical convergence guarantees numerical stability under small $\epsilon$, and therefore well defined both the theory and practice.
>
> # Rigorous Definition (Stein & Shakarchi)
> This visual intuition is formalized in Stein & Shakarchi’s *Functional Analysis* (Chapter 3, "Distributions: Generalized Functions"). Specifically, while $u$ has no classical derivative, it possesses a distributional derivative defined by the limit of the smooth sequence shown in Panel B. The derivative is defined by its action on test functions via integration by parts (Chapter 3.1, Eq 1). Physically, for a piecewise constant function, this derivative is a measure supported on the jump set--precisely what is visualized in Panel C.
>
> # Connection to Hodge Isomorphism
> Finally, **Panel (D)** demonstrates why this definition is mathematically powerful. One can integrate the gradient field from Panel C along a path $\gamma$ enclosing the domain. The plot tracks the accumulated rotation of the vectors. Despite the derivative being supported on a singular set (the boundary), the integral yields a clean integer value (winding number $\approx 1$). This illustrates the **Hodge Isomorphism Theorem**: the distributional derivative $du$ acts as a valid generator for the first cohomology group $H^1$, allowing us to rigorously detect topological features (holes) using the geometry of the boundary.
>
> [6] Functional Analysis: Introduction to Further Topics in Analysis., Stein, E. M., & Shakarchi, R., 2011

---

### Official Review · Reviewer_GGPn · 2025-10-31

**Soundness:** 3
**Presentation:** 3
**Contribution:** 3
**Rating:** 6
**Confidence:** 3

**Summary:**

This paper proposes an approach that guarantees a closed, simply connected mesh reconstruction from a noisy point cloud. It is inspired by the winding number field. The method takes a collection of oriented triangles of a 3D triangulation as input and outputs a closed, simply connected mesh. Besides, it obtains oriented triangles from a tetrahedral mesh and removes input outliers via alternating optimization. Experimental results on real and synthetic benchmark datasets demonstrate the effectiveness of the
approach.

**Strengths:**

This paper is well written with good presentation.

The implementation details have been elaborated clearly. It might be easy to reproduce the results reported in the paper.

The formulations are defined and explained clearly.

The improvements over prior methods are promising and remarkable.

**Weaknesses:**

It is important to clarify the new contributions more clearly, as the prooposed method is built on several existing methods like (Jacobson et 2023, Feng et al 2023). As claimed in the paper, the novelty is generalizing the existing framework from 2D curves to 3D surfaces.

Apart from textual descriptions and formal equations, it is common to provide a figure which can introduce the overal pipeline in this method.

Some presentations need more refinement, such as the captions in Figure 3 and 4 should not be on top.

**Questions:**

The experiments lack a comparison on the computation cost.

In terms of ablation study, it is important to provide more quantitative results apart from the comparison in Figure 5.

According to the quantitative comparisons in Table 1, why other competitors perform much poorly than the proposed method. Whether the implementation setups are fair properly?

---

> ### Author Response · Authors · 2025-11-25
> **Response to reviewer GGPn**
>
> We thank the reviewer for the positive assessment, particularly regarding the reproducibility, clarity of implementation, and the remarkable improvements over prior methods. We appreciate the constructive feedback regarding the distinction from prior work and the request for additional quantitative comparisons. We will update the manuscript with the clarifications below.
>
> # Clarification of Contributions
>
> While our work builds upon the winding number framework, it introduces three distinct technical contributions that are essential for the unorganized 3D point cloud setting, distinguishing it from Feng et al. (2023) and Jacobson et al. (2013).
>
> First, we generalize the domain from 2D curves/surfaces to 3D volumes. This is not merely a dimensional increase but requires a fundamental shift in the discrete differential operators. As detailed in Appendix A, we formulate the problem using volumetric discrete exterior derivative operators on a tetrahedral mesh to compute the Helmholtz-Hodge Decomposition of the 1-form $\omega$. This allows us to strictly enforce the simply connected topology of the level set surface, a guarantee not provided by standard surface-based cohomology processing.
>
> Second, unlike the 2D case where input edge directions are often known, unorganized point clouds lack consistent orientation. We introduce a novel Spectral Orientation method (Sec. 5.2) to solve this unsupervised global consistency problem. By analyzing the smallest eigenvectors of the operator $A^T A$ (derived from the HHD constraints), we resolve the orientation of triangles without ground truth supervision, a contribution absent in prior winding number literature.
>
> Third, regarding the optimization for topological error correction, Feng et al. (2023) relied on an expensive $L^1$ optimization (Integer Linear Programming) to produce sparse solutions for connecting holes. In contrast, we formulate an $L^0$ optimization problem (Eq. 8-9) that we solve via highly efficient Iteratively Reweighted Least Squares (IRLS). This approach not only promotes better sparsity and robustness against outliers--critical for defining the correction 2-chain $\Lambda$--but also enables significantly faster computation suitable for large-scale 3D inputs.
>
> # Pipeline Overview Figure
>
> We will enhance the caption of Figure 2 to explicitly map the visual steps to the corresponding algorithmic sections (Sec. 5 and Sec. 6).
>
> # Computational Cost Analysis
>
> We have compiled a detailed runtime breakdown on the CrossSDF dataset using an NVIDIA GH200 GPU. Our method is orders of magnitude faster than optimization-based neural baselines and significantly faster than CPU-based geometric methods.
>
> | Method | Screened PSR (CPU) | Neural-IMLS (GH200) | SALD (GH200) | Ours (GH200) |
> | :--- | :--- | :--- | :--- | :--- |
> | **Total Time** | 2.2s | 51 min | 37 min | **~0.53s** |
>
> Our speed is achieved through two highly optimized solver stages. First, the spectral orientation uses a Lanczos solver which converges in approximately 120 iterations. As noted in Sec. 7.1, a single iteration costs only 3.8 milliseconds for 100k points, resulting in a total time of roughly 500 milliseconds for this stage. Second, the final correction 2-chain is solved via our efficient Iteratively Reweighted Least Squares formulation. This boils down to 2-20 iterations of Geometric Multigrid (GMG) preconditioned Conjugate Gradient (CG). The CG iterations grow only marginally as the IRLS converges, with the total GPU solver cost for this step being approximately 19 milliseconds.
>
> # Quantitative Ablation Study
>
> To supplement the visual results in Figure 5, we provide the following quantitative table. It demonstrates that without specific components, the method fails to produce a valid mesh or suffers from low geometric accuracy.
>
> | Configuration | CD ($\times 100$) $\downarrow$ | HD ($\times 100$) $\downarrow$ | Success Rate (%) |
> | :--- | :--- | :--- | :--- |
> | **Full Method** | **0.26** | **4.3** | **100%** |
> | No Init. Orientation | 15.4 | 45.2 | 0% (Failed topology) |
> | No Robust ($L^0$) | 0.89 | 12.1 | 100% |
> | No Alternating | 1.12 | 18.5 | 100% |
>
> # Baseline Performance & Fairness
>
> The setup follows the standard CrossSDF protocol. The performance gap in Table 1 is primarily topological. Implicit methods (Neural-IMLS, POCO, PSR) attempt to fit smooth functions to noisy, thin structures. When noise exceeds the thinness of the structure (e.g., vessel walls), these methods "bridge" gaps or break connections, creating disconnected components (High CC). In contrast, our method is topologically correct by construction. We algebraically enforce a simply connected domain (CC=1), acting as a powerful regularizer that prevents fragmentation even in the presence of noise.

---

### Official Review · Reviewer_y33m · 2025-11-02

**Soundness:** 3
**Presentation:** 4
**Contribution:** 4
**Rating:** 8
**Confidence:** 2

**Summary:**

The paper presents a method to reconstruct closed, simply connected 3D meshes from noisy point clouds by first computing a 3D Delaunay triangulation and assigning triangle orientations via a spectral method, then iteratively applying a winding-number–based module to correct topology and refine the mesh. Experiments on medical and synthetic datasets show it produces topologically correct reconstructions with high geometric fidelity, outperforming existing methods

**Strengths:**

- The proposed method guarantees that the final 3D mesh is free of holes (topologically equivalent to a 2-sphere i.e., simply connected).
- It is a non–deep-learning approach that runs very efficiently and shows strong robustness to noisy point clouds, outperforming previous methods.
- The paper is exceptionally well presented, with clear overview sections that make the core ideas accessible even to readers outside this research area.

**Weaknesses:**

- The ablation study lacks quantitative results. The authors only include a qualitative 2D example (Figure 5) to demonstrate the importance of each component. It would strengthen the paper to report numerical metrics on full 3D models, similar to Tables 1 and 2.
- The evaluation metrics are described only briefly, authors can further elaborate why these metrics are suitable. In addition, highlighting the best metric values in bold would make the tables clearer and easier to interpret. It would help the reader if the best results in the tables were highlighted (e.g. in bold).
- It would be beneficial to include runtime comparison of the methods in Tables 1 & 2.

**Questions:**

- Is it guaranteed that this procedure will always converge?
- In line 179: Is it assumed that $f \in \mathcal{F}_0$ ? I haven't seen this to be explicitly stated, but I was wondering if it is not true, then (in line 183) shouldn't be $\Gamma_f \in \{ -1, 0, 1 \}$ ?
- Is there any reason for not including results on thick structures (as defined in the CrossSDF paper)?

---

> ### Author Response · Authors · 2025-11-25
> **Response to reviewer y33m**
>
> We sincerely thank the reviewer for the excellent rating and for highlighting our method's key strengths: the **topological guarantee** of simply connected reconstruction, **efficiency**, and **robustness** in an unsupervised manner. We are glad the presentation made the core concepts accessible. Below, we provide the additional quantitative data and clarifications requested to further strengthen the manuscript.
>
> # Quantitative Ablation Study
>
> To supplement the visual results in Figure 5, we provide the following quantitative table. It demonstrates that without specific components, the method fails to produce a valid mesh or suffers from low geometric accuracy.
>
> | Configuration | CD ($\times 100$) $\downarrow$ | HD ($\times 100$) $\downarrow$ | Success Rate (%) |
> | :--- | :--- | :--- | :--- |
> | **Full Method** | **0.26** | **4.3** | **100%** |
> | No Init. Orientation | 15.4 | 45.2 | 0% (Failed topology) |
> | No Robust ($L^0$) | 0.89 | 12.1 | 100% |
> | No Alternating | 1.12 | 18.5 | 100% |
>
> The "No Init. Orientation" configuration fails to recover the shape structure entirely, leading to high errors. The "No Robust ($L^0$)" configuration replaces the $L^0$ sparsity term with $L^2$; while it produces a valid topology, the surface is overly smooth and fails to snap to outliers, increasing Hausdorff Distance (HD). Finally, the "No Alternating" configuration fails to remove outliers effectively, resulting in visible geometric artifacts.
>
> # Computational Cost Comparison
>
> As requested, we have compiled a runtime comparison using the CrossSDF dataset on an NVIDIA GH200 GPU. Our method is orders of magnitude faster than optimization-based neural methods and significantly faster than the CPU-based Screened PSR. The table below represents the performance for a typical reconstruction of a **100k point scene**.
>
> | Method | Screened PSR (CPU) | Neural-IMLS (GH200) | SALD (GH200) | Ours (GH200) |
> | :--- | :--- | :--- | :--- | :--- |
> | **Total Time** | 2.2s | 51 min | 37 min | **~0.53s** |
>
> Our speed is achieved through two highly optimized solver stages. First, regarding the **Spectral Orientation**, a single iteration costs 3.8 milliseconds for 100k points as noted in Sec 7.1. Our method converges in approximately 120 iterations. Consequently, the total Lanczos GPU solver cost is roughly **500 milliseconds**.
>
> Second, regarding the **Correction 2-chain ($\Lambda$)**, this step is solved through highly efficient Iteratively Reweighted Least Squares (IRLS). This boils down to 2-20 iterations of Geometric Multigrid (GMG) Conjugate Gradient (CG). While CG iterations grow slightly as the IRLS converges, our GPU CG solver costs only **~19 milliseconds** in total for this step.
>
> # Response to Specific Questions
>
> **Clarification on Line 179:** Indeed, we assume $f \in \mathcal{F}_0$ in that context. We appreciate the reviewer for noticing this detail. We will revise the manuscript to make the notation explicitly consistent, clarifying that $\Gamma_f \in \{-1, 1\}$ for active faces in $\mathcal{F}_0$ and $\Gamma_f=0$ otherwise.
>
> **Convergence Guarantee:** The core numerical components of our algorithm--the Lanczos solver for spectral initialization and the Conjugate Gradient solver for linear systems--rely on symmetric positive-definite operators. These solvers have well-established theoretical convergence guarantees. In future work, we aim to further analyze the robustness of our spectral orientation solver through the **spectral gap** perspective. Specifically, quantitatively analyzing the spectral gap relative to input noise would establish precise noise tolerance levels while guaranteeing convergence.
>
> **Results on Thick Structures:** We prioritized "thin structures" because they represent the failure cases for most existing implicit and learning-based methods, which tend to bridge gaps or break thin connections. Thick structures are generally well-handled by standard baselines (like Screened PSR). Our method performs equally well on thick structures (recovering the exact topology and geometry), but we focused the evaluation on thin structures to highlight the specific advantage of our topological guarantees in challenging scenarios.
>
> # Metrics & Formatting
>
> We confirm that Chamfer Distance (CD) and Hausdorff Distance (HD) are used as standard metrics for geometric fidelity, while Connected Components (CC) serves as the critical proxy for topological correctness (a simply connected surface should have exactly 1 component). We will update Tables 1 and 2 to **bold** the best results for clearer interpretation.

---

### Meta-Review · Area_Chair_zXqc · 2026-01-06

**Summary:**

The primary concerns raised by reviewers centered on the paper's suitability for the ICLR venue, the clarity and novelty of its contributions relative to prior work, and the completeness of its empirical validation. Specifically, questions were raised about whether the advanced geometric and topological mathematics aligned with the conference's machine learning focus, and whether the claimed theoretical guarantees were presented with sufficient rigor. Requests were made for quantitative ablation studies, runtime comparisons, and a more precise discussion of limitations. After reviewing the authors' comprehensive rebuttal and the subsequent discussion, these concerns have been adequately addressed, leading to a final decision to accept the paper.

**Reviewer Concerns:**

The rebuttal effectively addressed the majority of concrete concerns. The authors provided a detailed quantitative ablation study and runtime analysis, clarified the mathematical foundations (including the distributional derivative and Hodge Isomorphism Theorem), and precisely differentiated their volumetric, optimization-based approach from prior winding number and topological surgery methods. The outstanding concern regarding venue fit, raised most prominently by Reviewer 9oMt, was mitigated through the discussion. The reviewer conceded that the work could be valuable to the learning community, and the authors successfully argued that their core spectral formulation and iterative optimization procedure fall within the broad scope of machine learning techniques. While the paper remains deeply technical, its "correct-by-construction" paradigm offers a valuable foundation for future learning-based methods.

**Reviewer Scores:**

Reviewer jxjb will increase the score.

---

### Decision · Program_Chairs · 2026-01-26

Accept (Poster)